# Distribution of Bipartite and Tripartite Entanglement within a Spin-1/2 Heisenberg Star in a Magnetic Field

**DOI:** 10.3390/molecules28104037

**Published:** 2023-05-11

**Authors:** Katarína Karlόvá, Jozef Strečka

**Affiliations:** Department of Theoretical Physics and Astrophysics, Faculty of Science, Pavol Jozef Šafárik University in Košice, Park Angelinum 9, 040 01 Košice, Slovakia; katarina.karlova@upjs.sk

**Keywords:** Heisenberg star, bipartite and tripartite entanglement, negativity, W-state

## Abstract

The spatial distribution of entanglement within a spin-1/2 Heisenberg star composed from a single central spin and three peripheral spins is examined in the presence of an external magnetic field using the Kambe projection method, which allows an exact calculation of the bipartite and tripartite negativity serving as a measure of the bipartite and tripartite entanglement. Apart from a fully separable polarized ground state emergent at high-enough magnetic fields, the spin-1/2 Heisenberg star exhibits at lower magnetic fields three outstanding nonseparable ground states. The first quantum ground state exhibits the bipartite and tripartite entanglement over all possible decompositions of the spin star into any pair or triad of spins, whereby the bipartite and tripartite entanglement between the central and peripheral spins dominates over that between the peripheral spins. The second quantum ground state has a remarkably strong tripartite entanglement between any triad of spins in spite of the lack of bipartite entanglement. The central spin of the spin star is separable from the remaining three peripheral spins within the third quantum ground state, where the peripheral spins are subject to the strongest tripartite entanglement arising from a two-fold degenerate W-state.

## 1. Introduction

Over the past few decades, molecular magnets have attracted a great deal of attention, because they provide a simple platform to encode a molecular spin qubit that could serve as a basic building block of novel quantum technologies [1]. The molecular spin qubit encoded in a single magnetic molecule can be coherently manipulated by the pulsed electron spin resonance, which is capable of controlling a state of the molecular spin qubit via a small oscillating magnetic field that rotates in a plane oriented perpendicular with respect to the applied time-independent magnetic field [2,3]. Single-molecule magnets displaying a magnetic hysteresis with rather long relaxation times have a great application potential for building extremely dense and efficient memory devices [4], which additionally allow the implementation of Grover’s search algorithm [5] by a multi-frequency sequence of electromagnetic pulses following the protocol due to Leuenberger and Loss [6]. Moreover, the exchange-coupled magnetic molecules afford a suitable resource for the implementation of two-qubit quantum gates [7]. The quantum entanglement between the molecular spin qubits may thus eventually provide a new route to quantum computation based on Shor’s factoring algorithm [8].

Altogether, it could be concluded that the quantum entanglement emergent in solid-state molecular systems affords a useful resource for quantum computation and the storing and processing of quantum information [9,10,11]. The strongest quantum entanglement can be generally expected in molecular antiferromagnets, whose magnetic properties are well-captured by the quantum Heisenberg spin model [12]. In the present article our particular attention will be focused on the bipartite and tripartite entanglement of the spin-1/2 Heisenberg star, which consists from a central spin interacting with three peripheral spins, as schematically illustrated in Figure 1. The ground state, magnetic and thermodynamic properties of the quantum Heisenberg spin star were comprehensively studied in the pioneering works by Richter and co-workers [13,14,15,16]. It is worthwhile to remark, moreover, that the quantum Heisenberg spin star is not just a theoretical curiosity without any connection to a real-world system, but it has a variety of experimental realizations in tetranuclear molecular complexes such as CrNi${}_{3}$ [17,18], CrMn${}_{3}$ [19], Cu${}_{4}$, Ni${}_{4}$ and NiCu${}_{3}$ [20]. From the perspective of quantum entanglement, only static and dynamic pairwise entanglement, two-point correlations and quantum discord of the spin-1/2 Heisenberg star with the exchange and Dzyaloshinskii–Moriya anisotropies were explored in detail in zero magnetic field and the absence of the exchange interaction between the peripheral spins [21,22].

The main goal of the present work is to clarify a spatial distribution of the bipartite and tripartite entanglement within the spin-1/2 Heisenberg star, which accounts for the exchange coupling between the central and peripheral spins, the exchange coupling between the peripheral spins, as well as the external magnetic field. To this end, we will rigorously calculate the bipartite and tripartite negativity [23,24,25,26,27] for all inequivalent decompositions of the spin-1/2 Heisenberg star into pairs or triads of spins. The main advantage of the quantity negativity with respect to other entanglement measures and witnesses lies in that it can be relatively simply calculated as a measure of both bipartite as well as multipartite entanglement [23,24,25,26,27]. The structure of the paper is organized as follows. In Section 2 we will introduce the investigated quantum spin model and clarify basic steps of the calculation procedure. The most interesting results for the measures of bipartite and tripartite entanglement are reported in Section 3. Finally, Section 4 provides a brief summary of the most important scientific findings. Some technical details concerned with the calculation procedure are given in Appendices Appendix A–Appendix D.

## 2. Model and Method

Let us consider the spin-1/2 Heisenberg star in a magnetic field, which is schematically illustrated in Figure 1 and given by the following Hamiltonian:(1)$$\begin{array}{c} \hat{H}=J{\hat{S}}_{0}\cdot \left({\hat{S}}_{1}+{\hat{S}}_{2}+{\hat{S}}_{3}\right)+J_{1}\left({\hat{S}}_{1}\cdot {\hat{S}}_{2}+{\hat{S}}_{2}\cdot {\hat{S}}_{3}+{\hat{S}}_{3}\cdot {\hat{S}}_{1}\right)-h\sum_{j=0}^{3}{\hat{S}}_{j}^{z}. \end{array}$$The coupling constant *J* determines the strength of the nearest-neighbor exchange interaction between the central spin $S_{0}$ and three peripheral spins $S_{1}$, $S_{2}$ and $S_{3}$, while the coupling constant $J_{1}$ determines the strength of the nearest-neighbor exchange interaction between the peripheral spins $S_{1}$, $S_{2}$ and $S_{3}$. An overall energy spectrum of the Hamiltonian (Equation 1) can be obtained with the Kambe projection method [28,29], which takes advantage of the validity of the commutation relations $\left[\hat{H},{\hat{S}}_{T}^{2}\right]=\left[\hat{H},{\hat{S}}_{T}^{z}\right]=\left[\hat{H},{\hat{S}}_{▵}^{2}\right]=0$ between the Hamiltonian (Equation 1) and the square of the total spin of the three peripheral spins ${\hat{S}}_{▵}={\hat{S}}_{1}+{\hat{S}}_{2}+{\hat{S}}_{3}$, the square of the total spin ${\hat{S}}_{T}={\hat{S}}_{0}+{\hat{S}}_{▵}$ and its *z*-component ${\hat{S}}_{T}^{z}={\hat{S}}_{0}^{z}+{\hat{S}}_{▵}^{z}$. The full energy spectrum of the spin-1/2 Heisenberg star in a magnetic field can be consequently expressed in terms of the corresponding quantum spin numbers $S_{T}$, $S_{▵}$ and $S_{T}^{z}$:(2)$$\begin{array}{c} E_{S_{T},S_{▵},S_{T}^{z}}=\frac{J}{2}\left[S_{T}\left(S_{T}+1\right)-S_{▵}\left(S_{▵}+1\right)-\frac{3}{4}\right]+\frac{J_{1}}{2}\left[S_{▵}\left(S_{▵+1}\right)-\frac{9}{4}\right]-hS_{T}^{z}. \end{array}$$All available combinations of the quantum spin numbers $S_{T}$, $S_{▵}$ and $S_{T}^{z}$ follow from basic quantum-mechanical rules $S_{T}=S_{▵}⊗S_{0}=\left(\frac{1}{2}⊗\frac{1}{2}⊗\frac{1}{2}\right)⊗\frac{1}{2}=\left(\frac{1}{2}⊕\frac{1}{2}⊕\frac{3}{2}\right)⊗\frac{1}{2}=0⊕1⊕0⊕1⊕1⊕2$ with the *z*-component of the total spin $S_{T}^{z}=-S_{T},-S_{T}+1,...,S_{T}$.

By solving the time-independent Schrödinger equation $\hat{H}|\psi_{i}\rangle =E_{i}|\psi_{i}\rangle$ one readily obtains all eigenvectors $|\psi_{i}\rangle =|S_{T},S_{▵},S_{T}^{z}\rangle$ of the spin-1/2 Heisenberg star in a magnetic field, which are explicitly listed in Table 1 together with the corresponding energy eigenvalues. From the full energy spectrum listed in Table 1 one may consequently calculate the partition function:(3)$$\begin{array}{ccc} Z\;\; & = & \;\;\sum_{i=1}^{16}\exp\left(\beta E_{i}\right)=2\exp\left(-\frac{3}{4}\beta J-\frac{3}{4}\beta J_{1}\right)\cosh\left(2\beta h\right)+\exp\left(-\frac{3}{4}\beta J-\frac{3}{4}\beta J_{1}\right)\cosh\left(\beta h\right)\\ \;\; & + & \;\;\exp\left(-\frac{3}{4}\beta J-\frac{3}{4}\beta J_{1}\right)+2\exp\left(\frac{5}{4}\beta J-\frac{3}{4}\beta J_{1}\right)\cosh\left(\beta h\right)+\exp\left(\frac{5}{4}\beta J-\frac{3}{4}\beta J_{1}\right)\\ \;\; & + & \;\;4\exp\left(-\frac{\beta J}{4}+\frac{3}{4}\beta J_{1}\right)\cosh\left(\beta h\right)+2\exp\left(-\frac{\beta J}{4}+\frac{3}{4}\beta J_{1}\right)+2\exp\left(\frac{3}{4}\beta J+\frac{3}{4}\beta J_{1}\right). \end{array}$$

### 2.1. Bipartite Entanglement

For a quantification of the degree of bipartite entanglement in the spin-1/2 Heisenberg star we will adapt the quantity negativity introduced according to the Peres–Horodecki concept [23,24]. Unlike the original definition put forward by Vidal and Werner [25], we will henceforth employ the alternate definition of the negativity with twice as large a value [26,27]. It should be stressed, moreover, that one may calculate two different measures of the bipartite entanglement within the spin-1/2 Heisenberg star by considering all available decompositions of the spin star into spin pairs. Namely, the negativity $N_{01}$ will measure the bipartite entanglement between the central spin $S_{0}$ and one of the peripheral spins (e.g., $S_{1}$), while the negativity $N_{12}$ will measure the bipartite entanglement between two peripheral spins (e.g., $S_{1}$ and $S_{2}$).

The starting point for the calculation of both bipartite negativities $N_{01}$ and $N_{12}$ is the evaluation of the overall density operator, which can be put into a more convenient form for subsequent calculations using the spectral decomposition into orthogonal projections including the complete set of eigenvectors given in Table 1:(4)$$\begin{array}{c} \hat{\rho }=\frac{1}{Z}\exp\left(-\beta \hat{H}\right)=\frac{1}{Z}\sum_{n=1}^{16}\exp\left(-\beta E_{n}\right)|\psi_{n}\rangle \langle \psi_{n}|. \end{array}$$To evaluate the bipartite negativity for some general spin pair $S_{i}$-$S_{j}$ one should first calculate the relevant reduced density operator ${\hat{\rho }}^{ij}$ by tracing out the degrees of freedom of the remaining two spins $S_{k}$ and $S_{l}$ of the spin-1/2 Heisenberg star:(5)$$\begin{array}{c} {\hat{\rho }}^{ij}={\mathrm{Tr}}_{S_{k}}{\mathrm{Tr}}_{S_{l}}\hat{\rho }=\frac{1}{Z}\sum_{n=1}^{16}\sum_{S_{k}^{z}=\pm 1/2}\sum_{S_{l}^{z}=\pm 1/2}\exp\left(-\beta E_{n}\right)\langle S_{k}^{z},S_{l}^{z}|\psi_{n}\rangle \langle \psi_{n}|S_{k}^{z},S_{l}^{z}\rangle . \end{array}$$Hence, the negativity $N_{01}$ measuring the strength of the bipartite entanglement between the central spin $S_{0}$ and the peripheral spin $S_{1}$ can be computed from the reduced density operator ${\hat{\rho }}^{01}$ obtained after tracing out the degrees of freedom of the peripheral spins $S_{2}$ and $S_{3}$, while the negativity $N_{12}$ measuring the strength of the bipartite entanglement between two peripheral spins $S_{1}$ and $S_{2}$ can be calculated from the reduced density operator ${\hat{\rho }}^{12}$ obtained after tracing out the degrees of freedom of the central spin $S_{0}$ and the peripheral spin $S_{3}$. Both the aforementioned measures of the bipartite entanglement $N_{01}$ and $N_{12}$ can thus be obtained from the formally same reduced density matrix:(6)$$\begin{array}{c} \rho^{ij}=\left(\;\;\begin{array}{cccc} \rho_{11}^{ij} & 0 & 0 & 0\\ 0 & \rho_{22}^{ij} & \rho_{23}^{ij} & 0\\ 0 & \rho_{32}^{ij} & \rho_{33}^{ij} & 0\\ 0 & 0 & 0 & \rho_{44}^{ij} \end{array}\;\;\right), \end{array}$$
which in fact represents a matrix representation of the reduced density operator (Equation 5) in the standard basis of the two remaining spins ${|\;↑}_{i}{↑}_{j}\rangle$, ${|\;↑}_{i}{↓}_{j}\rangle$, ${|\;↓}_{i}{↑}_{j}\rangle$, ${|\;↓}_{i}{↓}_{j}\rangle$. The only difference between the density matrices $\rho^{01}$ and $\rho^{12}$ lies in an explicit form of their elements, which are for completeness explicitly listed in Appendices Appendix A and Appendix B.

In order to proceed further with the calculation of the bipartite negativity for the spins $S_{i}$ and $S_{j}$ one should consecutively perform a partial transposition of the reduced density matrix $\rho^{ij}$ with respect to either the spin $S_{i}$ or $S_{j}$. The partial transposition $T_{j}$ with respect to the spin $S_{j}$ affords the partially transposed reduced density matrix:(7)$$\begin{array}{c} {\left(\rho^{ij}\right)}^{T_{j}}=\left(\;\;\begin{array}{cccc} \rho_{11}^{ij} & 0 & 0 & \rho_{23}^{ij}\\ 0 & \rho_{22}^{ij} & 0 & 0\\ 0 & 0 & \rho_{33}^{ij} & 0\\ \rho_{32}^{ij} & 0 & 0 & \rho_{44}^{ij} \end{array}\;\;\right). \end{array}$$After diagonalizing the partially transposed reduced density matrix (Equation 7) one acquires the following four eigenvalues:(8)$$\begin{array}{ccc} \lambda_{1,2}^{ij}\;\; & = & \;\;\frac{1}{2}\left[\rho_{11}^{ij}+\rho_{44}^{ij}\pm \sqrt{{\left(\rho_{11}^{ij}-\rho_{44}^{ij}\right)}^{2}+4{\left(\rho_{23}^{ij}\right)}^{2}}\right],\\ \lambda_{3}^{ij}\;\; & = & \;\;\rho_{22}^{ij},\;\;\lambda_{4}^{ij}=\rho_{33}^{ij}, \end{array}$$
among which only the eigenvalue with a minus sign in front of the square root may become negative. According to the Peres–Horodecki separability criterion [23,24], the necessary and sufficient condition for the presence of quantum entanglement is at least one negative eigenvalue of the partially transposed reduced density matrix. The quantity negativity, which refers to the sum of the absolute values of the negative eigenvalues of the partially transposed reduced density matrix, can be accordingly considered as a quantitative measure of the bipartite entanglement [25,26,27]:(9)$$\begin{array}{c} N_{ij}=\max\left\{0,\sum_{n=1}^{4}\left(\right|\lambda_{n}^{ij}\left|-\lambda_{n}^{ij}\right)\right\}. \end{array}$$The negativities $N_{01}$ and $N_{12}$ measuring a strength of the bipartite entanglement in the spin-1/2 Heisenberg star are consequently given by the formula:(10)$$\begin{array}{c} N_{ij}=\max\left\{0,\sqrt{{\left(\rho_{11}^{ij}-\rho_{44}^{ij}\right)}^{2}+4{\left(\rho_{23}^{ij}\right)}^{2}}-\left(\rho_{11}^{ij}+\rho_{44}^{ij}\right)\right\}. \end{array}$$Substituting into Equation (Equation 10) the respective elements of the reduced density matrix $\rho^{01}$ ($\rho^{12}$) listed in Appendixes Appendix A and Appendix B, one obtains the bipartite negativity $N_{01}$ ($N_{12}$) calculated for the central spin $S_{0}$ and the peripheral spin $S_{1}$ (the peripheral spins $S_{1}$ and $S_{2}$).

### 2.2. Tripartite Entanglement

It is noteworthy that the absence of the bipartite entanglement does not generally exclude multiparticle entanglement. The tripartite negativity represents a useful measure of the tripartite entanglement, which allows one to discriminate fully separable or biseparable states from tripartite entangled states [30,31]. The tripartite negativity quantifying a degree of the tripartite entanglement between the spins $S_{i}$, $S_{j}$ and $S_{k}$ can be defined as the geometric mean of three bipartite negativities [30]:(11)$$\begin{array}{c} N_{ijk}=\sqrt[{3}]{N_{i-jk}N_{j-ik}N_{k-ij}}. \end{array}$$The bipartite negativity $N_{i-jk}$ measures the degree of bipartite entanglement between the spin $S_{i}$ and the spin pair $S_{j}-S_{k}$, which can be calculated from eigenvalues of the reduced density matrix partially transposed with respect to the spin $S_{i}$. To this end, it is necessary to calculate the reduced density operator ${\hat{\rho }}^{ijk}$ for the spins $S_{i}$, $S_{j}$ and $S_{k}$ by tracing out degrees of freedom of the fourth spin $S_{l}$ from the overall density operator (Equation 4):(12)$$\begin{array}{c} {\hat{\rho }}^{ijk}={\mathrm{Tr}}_{S_{l}}\hat{\rho }=\frac{1}{Z}\sum_{n=1}^{16}\sum_{S_{l}^{z}=\pm 1/2}\exp\left(-\beta E_{n}\right)\langle S_{l}^{z}|\psi_{n}\rangle \langle \psi_{n}|S_{l}^{z}\rangle . \end{array}$$The matrix representation of the reduced density operator (Equation 12) in the standard spin basis ${|\;↑}_{i}{↑}_{j}{↑}_{k}\rangle$, ${|\;↑}_{i}{↑}_{j}{↓}_{k}\rangle$, ${|\;↑}_{i}{↓}_{j}{↑}_{k}\rangle$,${|\;↑}_{i}{↓}_{j}{↓}_{k}\rangle$, ${|\;↓}_{i}{↑}_{j}{↑}_{k}\rangle$, ${|\;↓}_{i}{↑}_{j}{↓}_{k}\rangle$, ${|\;↓}_{i}{↓}_{j}{↑}_{k}\rangle$, ${|\;↓}_{i}{↓}_{j}{↓}_{k}\rangle$ is given by:(13)$$\begin{array}{c} \rho^{ijk}=\left(\;\;\begin{array}{cccccccc} \rho_{11}^{ijk} & 0 & 0 & 0 & 0 & 0 & 0 & 0\\ 0 & \rho_{22}^{ijk} & \rho_{23}^{ijk} & 0 & \rho_{25}^{ijk} & 0 & 0 & 0\\ 0 & \rho_{23}^{ijk} & \rho_{22}^{ijk} & 0 & \rho_{25}^{ijk} & 0 & 0 & 0\\ 0 & 0 & 0 & \rho_{44}^{ijk} & 0 & \rho_{46}^{ijk} & \rho_{46}^{ijk} & 0\\ 0 & \rho_{25}^{ijk} & \rho_{25}^{ijk} & 0 & \rho_{55}^{ijk} & 0 & 0 & 0\\ 0 & 0 & 0 & \rho_{46}^{ijk} & 0 & \rho_{66}^{ijk} & \rho_{67}^{ijk} & 0\\ 0 & 0 & 0 & \rho_{46}^{ijk} & 0 & \rho_{67}^{ijk} & \rho_{66}^{ijk} & 0\\ 0 & 0 & 0 & 0 & 0 & 0 & 0 & \rho_{88}^{ijk} \end{array}\;\;\right). \end{array}$$It is quite obvious that the individual elements of the reduced density matrix (Equation 13) will basically depend on whether one traces out in Equation (Equation 12) the degrees of freedom of the central spin $S_{0}$ in order to obtain the density matrix $\rho^{123}$ or traces out in Equation (Equation 12) the degrees of freedom of one peripheral spin $S_{3}$ in order to obtain the density matrix $\rho^{012}$. The individual elements of the reduced density matrices $\rho^{012}$ and $\rho^{123}$ are for the sake of completeness explicitly quoted in Appendices Appendix C and Appendix D, respectively.

Next, one may perform a partial transposition of the reduced density matrix (Equation 13) with respect to the spin $S_{i}$ in order to obtain the partially transposed reduced density matrix:(14)$$\begin{array}{c} \rho^{i-jk}={\left(\rho^{ijk}\right)}^{T_{i}}=\left(\;\;\begin{array}{cccccccc} \rho_{11}^{ijk} & 0 & 0 & 0 & 0 & \rho_{25}^{ijk} & \rho_{25}^{ijk} & 0\\ 0 & \rho_{22}^{ijk} & \rho_{23}^{ijk} & 0 & 0 & 0 & 0 & \rho_{46}^{ijk}\\ 0 & \rho_{23}^{ijk} & \rho_{22}^{ijk} & 0 & 0 & 0 & 0 & \rho_{46}^{ijk}\\ 0 & 0 & 0 & \rho_{44}^{ijk} & 0 & 0 & 0 & 0\\ 0 & 0 & 0 & 0 & \rho_{55}^{ijk} & 0 & 0 & 0\\ \rho_{25}^{ijk} & 0 & 0 & 0 & 0 & \rho_{66}^{ijk} & \rho_{67}^{ijk} & 0\\ \rho_{25}^{ijk} & 0 & 0 & 0 & 0 & \rho_{67}^{ijk} & \rho_{66}^{ijk} & 0\\ 0 & \rho_{46}^{ijk} & \rho_{46}^{ijk} & 0 & 0 & 0 & 0 & \rho_{88}^{ijk} \end{array}\;\;\right). \end{array}$$Owing to the higher symmetry, the eigenvalues of the partially transposed reduced density matrices $\rho^{0-12}={\left(\rho^{012}\right)}^{T_{0}}$, $\rho^{1-23}={\left(\rho^{123}\right)}^{T_{1}}$, $\rho^{2-13}={\left(\rho^{123}\right)}^{T_{2}}$, and $\rho^{3-12}={\left(\rho^{123}\right)}^{T_{3}}$ are given by the relatively simple expressions:(15)$$\begin{array}{ccc} \lambda_{1}^{i-jk}\;\;\; & = & \;\;\;\rho_{44}^{ijk},\;\lambda_{2}^{i-jk}=\rho_{55}^{ijk},\\ \lambda_{3}^{i-jk}\;\;\; & = & \;\;\;\rho_{44}^{ijk}-\rho_{46}^{ijk},\\ \lambda_{4,5}^{i-jk}\;\;\; & = & \;\;\;\frac{1}{2}\left[\rho_{11}^{ijk}+\rho_{44}^{ijk}+\rho_{46}^{ijk}\pm \sqrt{{\left(\rho_{11}^{ijk}-\rho_{44}^{ijk}-\rho_{46}^{ijk}\right)}^{2}+8{\left(\rho_{23}^{ijk}\right)}^{2}}\right],\\ \lambda_{6}^{i-jk}\;\;\; & = & \;\;\;\rho_{22}^{ijk}-\rho_{23}^{ijk},\\ \lambda_{7,8}^{i-jk}\;\;\; & = & \;\;\;\frac{1}{2}\left[\rho_{88}^{ijk}+\rho_{22}^{ijk}+\rho_{23}^{ijk}\pm \sqrt{{\left(\rho_{88}^{ijk}-\rho_{22}^{ijk}-\rho_{23}^{ijk}\right)}^{2}+8{\left(\rho_{46}^{ijk}\right)}^{2}}\right]. \end{array}$$

Let us further perform a partial transposition of the reduced density matrix $\rho^{1-02}\;=\;\rho^{2-01}$,
(16)$$\begin{array}{c} \rho^{1-02}={\left(\rho^{012}\right)}^{T_{1}}=\left(\;\;\begin{array}{cccccccc} \rho_{11}^{012} & 0 & 0 & \rho_{23}^{012} & 0 & 0 & \rho_{25}^{012} & 0\\ 0 & \rho_{22}^{012} & 0 & 0 & \rho_{25}^{012} & 0 & 0 & \rho_{46}^{012}\\ 0 & 0 & \rho_{22}^{012} & 0 & 0 & 0 & 0 & 0\\ \rho_{23}^{012} & 0 & 0 & \rho_{44}^{012} & 0 & 0 & \rho_{46}^{012} & 0\\ 0 & \rho_{25}^{012} & 0 & 0 & \rho_{55}^{012} & 0 & 0 & \rho_{67}^{012}\\ 0 & 0 & 0 & 0 & 0 & \rho_{66}^{012} & 0 & 0\\ \rho_{25}^{012} & 0 & 0 & \rho_{46}^{012} & 0 & 0 & \rho_{66}^{012} & 0\\ 0 & \rho_{46}^{012} & 0 & 0 & \rho_{67}^{012} & 0 & 0 & \rho_{88}^{012} \end{array}\;\;\right). \end{array}$$

On the other hand, the eigenvalues of two less-symmetric partially transposed density matrices $\rho^{1-02}={\left(\rho^{012}\right)}^{T_{1}}$ and $\rho^{2-01}={\left(\rho^{012}\right)}^{T_{2}}$ are given by more complicated expressions:(17)$$\begin{array}{ccc} \lambda_{j}^{1-02}\;\;\; & = & \;\;\;\lambda_{j}^{2-01}=\frac{a_{1}}{3}+2\mathrm{sgn}\left(q_{1}\right)\sqrt{p_{1}}\cos\left\{\frac{1}{3}\left[\phi_{1}+\left(j-1\right)2\pi \right]\right\},\;j=1-3,\\ \lambda_{k}^{1-02}\;\;\; & = & \;\;\;\lambda_{k}^{2-01}=\frac{a_{2}}{3}+2\mathrm{sgn}\left(q_{2}\right)\sqrt{p_{2}}\cos\left\{\frac{1}{3}\left[\phi_{2}+\left(k-4\right)2\pi \right]\right\},\;k=4-6,\\ \lambda_{7}^{1-02}\;\;\; & = & \;\;\;\lambda_{7}^{2-01}=\rho_{22}^{012},\;\lambda_{8}^{1-02}=\lambda_{8}^{2-01}=\rho_{66}^{012}, \end{array}$$
whereby the coefficients entering into the relevant eigenvalues are defined as follows:(18)$$\begin{array}{ccc} p_{i}\;\;\; & = & \;\;\;\frac{a_{i}^{2}}{9}-\frac{b_{i}}{3},\;q_{i}=\frac{a_{i}^{3}}{27}-\frac{a_{i}b_{i}}{6}-\frac{c_{i}}{2},\;\phi_{i}=\arctan\left(\frac{\sqrt{p_{i}^{3}-q_{i}^{2}}}{q_{i}}\right);\;i=1-2,\\ a_{1}\;\;\;\; & = & \;\;\;\;\rho_{11}^{012}\;+\;\rho_{44}^{012}\;+\;\rho_{66}^{012};\;b_{1}\;\;=\;\;\rho_{11}^{012}\rho_{44}^{012}\;+\;\rho_{11}^{012}\rho_{66}^{012}\;+\;\rho_{44}^{012}\rho_{66}^{012}\;-\;{\left(\rho_{23}^{012}\right)}^{2}\;-\;{\left(\rho_{25}^{012}\right)}^{2}\;-\;{\left(\rho_{46}^{012}\right)}^{2},\\ c_{1}\;\;\;\; & = & \;\;\;\;\rho_{11}^{012}{\left(\rho_{46}^{012}\right)}^{2}\;+\;\rho_{44}^{012}{\left(\rho_{25}^{012}\right)}^{2}\;+\;\rho_{66}^{012}{\left(\rho_{23}^{012}\right)}^{2}\;-\;\rho_{11}^{012}\rho_{44}^{012}\rho_{66}^{012}\;-\;2\rho_{23}^{012}\rho_{25}^{012}\rho_{46}^{012},\\ a_{2}\;\;\;\; & = & \;\;\;\;\rho_{22}^{012}\;+\;\rho_{55}^{012}\;+\;\rho_{88}^{012};\;b_{1}\;\;=\;\;\rho_{22}^{012}\rho_{55}^{012}\;+\;\rho_{22}^{012}\rho_{88}^{012}\;+\;\rho_{55}^{012}\rho_{88}^{012}\;-\;{\left(\rho_{25}^{012}\right)}^{2}\;-\;{\left(\rho_{46}^{012}\right)}^{2}\;-\;{\left(\rho_{67}^{012}\right)}^{2},\\ c_{2}\;\;\;\; & = & \;\;\;\;\rho_{22}^{012}{\left(\rho_{67}^{012}\right)}^{2}+\rho_{55}^{012}{\left(\rho_{46}^{012}\right)}^{2}+\rho_{88}^{012}{\left(\rho_{25}^{012}\right)}^{2}-\rho_{22}^{012}\rho_{55}^{012}\rho_{88}^{012}-2\rho_{25}^{012}\rho_{46}^{012}\rho_{75}^{012}. \end{array}$$The bipartite negativity determining the strength of the bipartite entanglement between the spin $S_{i}$ and the spin pair $S_{j}-S_{k}$ can be finally calculated as the sum of the absolute values of the negative eigenvalues of the partially transposed reduced density matrix ${\left(\rho^{ijk}\right)}^{T_{i}}$ [26,27]:(19)$$\begin{array}{c} N_{i-jk}=\max\left\{0,\sum_{n=1}^{8}\left(\right|\lambda_{n}^{i-jk}\left|-\lambda_{n}^{i-jk}\right)\right\}. \end{array}$$The partially transposed reduced density matrices $\rho^{1-23}={\left(\rho^{123}\right)}^{T_{1}}$, $\rho^{2-13}={\left(\rho^{123}\right)}^{T_{2}}$ and $\rho^{3-12}={\left(\rho^{123}\right)}^{T_{3}}$ have, due to symmetry, the same set of eigenvalues (Equation 15), which immediately implies equality of the bipartite negativities $N_{1-23}=N_{2-13}=N_{3-12}$. The tripartite negativity $N_{123}$ calculated for the three peripheral spins $S_{1}$, $S_{2}$ and $S_{3}$ of the spin-1/2 Heisenberg star consequently satisfies the following simple formula:(20)$$\begin{array}{c} N_{123}=\sqrt[{3}]{N_{1-23}N_{2-13}N_{3-12}}=N_{1-23}. \end{array}$$Contrary to this, the eigenvalues (Equation 15) of the partially transposed reduced density matrix $\rho^{0-12}={\left(\rho^{012}\right)}^{T_{0}}$ generally differ from the eigenvalues (Equation 17) of the partially transposed reduced density matrices $\rho^{1-02}={\left(\rho^{012}\right)}^{T_{1}}$ and $\rho^{2-01}={\left(\rho^{012}\right)}^{T_{2}}$, which is consistent with inequality of the bipartite negativities $N_{0-12}\neq N_{1-02}=N_{2-01}$. Bearing this in mind, the tripartite negativity $N_{012}$ calculated for the central spin $S_{0}$ and two peripheral spins $S_{1}$ and $S_{2}$ of the spin-1/2 Heisenberg star should satisfy the formula:(21)$$\begin{array}{c} N_{012}=\sqrt[{3}]{N_{0-12}N_{1-02}N_{2-01}}=\sqrt[{3}]{N_{0-12}N_{1-02}^{2}}. \end{array}$$

## 3. Results and Discussion

Let us proceed to a discussion of the most interesting results for the bipartite and tripartite entanglement of the spin-1/2 Heisenberg star. The distribution of quantum entanglement in the spin-1/2 Heisenberg star can be inferred from the density plots of the bipartite and tripartite negativities depicted in Figure 2 in the interaction ratio $J_{1}/J$ versus magnetic field h/J plane serving as a sort of ground-state phase diagram. It follows from this figure that the ground-state phase diagram of the spin-1/2 Heisenberg star involves in total four different phases, which are unambiguously given in Figure 2 through their respective eigenvectors $|S_{T},S_{▵},S_{T}^{z}\rangle$ whose more explicit form is listed in Table 1. It is quite evident from Figure 2 that the bipartite and tripartite entanglement is completely absent in the fully separable polarized state |2,3/2,2〉 and our further attention will be therefore concentrated on the remaining three ground states |1,3/2,1〉, |1,1/2,1〉 and |0,1/2,0〉. While the former ground state |1,3/2,1〉 is unique (nondegenerate), the other two ground states |1,1/2,1〉 and |0,1/2,0〉 are twofold degenerate.

The ground state |1,3/2,1〉 can be characterized by the strongest bipartite entanglement between the central and peripheral spins among three nonseparable ground states. In fact, a rather strong bipartite entanglement can be found in the ground state |1,3/2,1〉 between the central and peripheral spins $N_{01}(|1,3/2,1\rangle )=\frac{1}{6}\left(\sqrt{10}-1\right)≐0.360$ (see Figure 2a), which is, however, accompanied by much weaker bipartite entanglement between the peripheral spins $N_{12}(|1,3/2,1\rangle )=\frac{1}{6}\left(\sqrt{26}-5\right)≐0.017$ (see Figure 2b). The ground state |1,1/2,1〉 displays the bipartite entanglement between the peripheral spins of relatively intense value $N_{12}(|1,1/2,1\rangle )=\frac{1}{3}\left(\sqrt{2}-1\right)≐0.138$, whereas this phase, contrarily, does not show any bipartite entanglement between the central and peripheral spin $N_{01}(|1,1/2,1\rangle )=0$. It is even more surprising that no bipartite entanglement has been detected within the ground state |0,1/2,0〉—neither between the central and peripheral spins $N_{01}(|0,1/2,0\rangle )=0$, nor between two peripheral spins $N_{12}(|0,1/2,0\rangle )=0$.

Bearing this in mind, it appears worthwhile to investigate a distribution of the tripartite entanglement within the spin-1/2 Heisenberg star. It turns out that the ground state |0,1/2,0〉 without the bipartite entanglement shows the same strength of the tripartite entanglement $N_{012}(|0,1/2,0\rangle )=N_{123}(|0,1/2,0\rangle )=\frac{1}{3}≐0.333$ between the central spin and two peripheral spins, as well as the three peripheral spins. Furthermore, the absence of tripartite entanglement between the central and peripheral spins $N_{012}(|1,1/2,1\rangle )=0$ within the ground state |1,1/2,1〉 is accompanied by a relatively strong tripartite entanglement between the peripheral spins $N_{123}(|1,1/2,1\rangle )=\frac{\sqrt{2}}{3}≐0.471$. In agreement with the expectations, the latter nonzero value of the tripartite negativity $N_{123}$ acquires exactly a half of the typical value for the W-state due to a two-fold degeneracy of the ground state |1,1/2,1〉 [30]. Finally, the tripartite entanglement within the ground state |1,3/2,1〉 bears a close relation to the bipartite one. The tripartite negativity is relatively high between the central and peripheral spins $N_{012}(|1,3/2,1\rangle )=\frac{1}{12}{\left(\sqrt{73}-1\right)}^{1/3}{\left(\sqrt{41}-1\right)}^{2/3}≐0.503$, while it becomes relatively small $N_{123}(|1,3/2,1\rangle )=\frac{1}{4}\left(\frac{\sqrt{89}}{3}-3\right)≐0.036$ between the peripheral spins within the ground state |1,3/2,1〉.

### 3.1. Thermal Bipartite Entanglement

Now, let us focus our attention on a detailed analysis of the bipartite entanglement at finite temperatures, which is traditionally referred to as the bipartite thermal entanglement. First, we will examine the bipartite thermal entanglement between the central and peripheral spins of the spin-1/2 Heisenberg star quantified by the bipartite negativity $N_{01}$. Four typical scans of the bipartite negativity $N_{01}$ across the ground-state phase diagram are plotted in Figure 3. The high values of the bipartite negativity $N_{01}$ are proliferated over the widest range of temperatures and magnetic fields for a sufficiently small value of the interaction ratio $J_{1}/J=0.25$, which promotes the bipartite entanglement between the central and peripheral spins within the ground state |1,3/2,1〉. As one could expect, the bipartite thermal entanglement of this type is gradually reduced upon an increasing in the magnetic field and temperature (see Figure 3a). In contrast, the high nonzero values of the bipartite negativity $N_{01}$ are for the moderate value of the interaction ratio $J_{1}/J=0.75$ limited to a dome-like parameter region, because the bipartite entanglement between the central and peripheral spins is restricted to moderate magnetic fields stabilizing the ground state |1,3/2,1〉 (see Figure 3b). The dome-like behavior in a moderate range of the magnetic fields still persists for the special case of the interaction ratio $J_{1}/J=1.0$, but the bipartite negativity $N_{01}$ acquires much smaller values due to a mixed state originating from the ground states |1,3/2,1〉 and |1,1/2,1〉 that coexist together (see Figure 3c). Although the zero-temperature bipartite negativity $N_{01}$ is zero at a higher value of the interaction ratio $J_{1}/J=1.25$ for an arbitrary magnetic field, it surprisingly turns out that a very weak bipartite thermal entanglement between the central and peripheral spins can be invoked at finite temperatures in proximity to the magnetic-field range h/J∈(2,3). The strongest bipartite thermal entanglement between the central and peripheral spins is strikingly concentrated close to a coexistence point of two ground states |1,1/2,1〉 and |2,3/2,2〉 with zero bipartite negativity $N_{01}$.

Next, let us proceed to a discussion of the bipartite thermal entanglement between two peripheral spins of the spin-1/2 Heisenberg star, which can be deduced from density plots of the bipartite negativity $N_{12}$ depicted in Figure 4 for four different values of the interaction ratio. It can be seen from Figure 4 that the density plot of the bipartite negativity $N_{12}$ always displays the dome-like shape regardless of the interaction ratio. If the interaction ratio is sufficiently small $J_{1}/J<1$ (Figure 4a,b) the relatively weak bipartite entanglement between the peripheral spins originates from the phase |1,3/2,1〉, whereas somewhat stronger bipartite entanglement between the peripheral spins results from the phase |1,1/2,1〉 for higher values of the interaction ratio $J_{1}/J>1$ (see Figure 4d). The most crucial difference between Figure 4a,b is the magnetic-field range where the bipartite negativity $N_{12}$ is nonzero, which extends either to zero magnetic field (Figure 4a) or some finite magnetic field (Figure 4b) in accordance with a stability condition of the ground state |1,3/2,1〉. For the particular case $J_{1}/J=1$ the ground states |1,3/2,1〉 and |1,1/2,1〉 coexist together in the magnetic-field range h/J∈(1;2) and hence, the bipartite negativity reaches the special value $N_{12}≐0.027$ that interpolates between the values ascribed to the ground states |1,3/2,1〉 and |1,1/2,1〉 (see Figure 4c). Another interesting observation is that the dome-like domain with the nonzero bipartite negativity $N_{12}$ is tilted towards higher magnetic fields upon an increase in the temperature, akin the leaning tower of Pisa, on the assumption that the interaction ratio $J_{1}/J\leq 1$ (see Figure 4a–c).

### 3.2. Thermal Tripartite Entanglement

Last but not least, we will proceed to a discussion of the tripartite thermal entanglement emergent within the spin-1/2 Heisenberg star in a magnetic field. First, our attention will be focused on the tripartite thermal entanglement between the central and two peripheral spins, which can be inferred from the density plots of the tripartite negativity $N_{012}$ shown in Figure 5. It can be seen from Figure 5a that the strongest tripartite entanglement between the central and two peripheral spins can be detected for the relatively small value of the interaction ratio $J_{1}/J=0.25$, which favors at low-enough magnetic fields the ground state |1,3/2,1〉 with the strongest bipartite and tripartite quantum correlations between the central and peripheral spins. The tripartite negativity $N_{012}$ also bears evidence of a peculiar magnetic-field-driven enhancement of the respective tripartite entanglement at a moderate value of the interaction ratio $J_{1}/J=0.75$, which relates to a magnetic-field-induced transition from a less-entangled ground state |0,1/2,0〉 to a more-entangled ground state |1,3/2,1〉 (Figure 5b). The opposite trend can be observed in Figure 5c for the particular value of the interaction ratio $J_{1}/J=1$, which has a much smaller value of the tripartite negativity $N_{012}$ in a range of the moderate magnetic fields h/J∈(1,2) due to a mixed state originating from a phase coexistence of the two ground states |1,3/2,1〉 and |1,1/2,1〉. Although the ground state |1,1/2,1〉 suffers from a lack of tripartite entanglement between the central and two peripheral spins, it follows from Figure 5d that a relatively weak tripartite thermal entanglement of this type can eventually be invoked above the ground state |1,1/2,1〉 with zero tripartite negativity $N_{012}=0$.

Let us conclude our analysis by investigating the tripartite entanglement between three peripheral spins of the spin-1/2 Heisenberg star in a magnetic field, which can be deduced from the density plots of the tripartite negativity $N_{123}$ displayed in Figure 6. It turns out that the tripartite entanglement between three peripheral spins exists at low-enough temperatures and magnetic fields irrespective of the interaction ratio. It can be easily understood that the size of the tripartite negativity $N_{123}$ generally enhances upon an increase in the interaction ratio due to strengthening of the pair correlations between the peripheral spins. In contrast to the previous case, the weakest tripartite entanglement between three peripheral spins can be thus detected for the smallest value of the interaction ratio $J_{1}/J=0.25$, which gives rise to the ground state |1,3/2,1〉 (see Figure 6a). At a moderate value of the interaction ratio $J_{1}/J=0.75$ one contrarily observes two pronounced dome-shaped domains with the nonzero tripartite negativity $N_{123}$ (see Figure 6b). The former dome shows the higher tripartite negativity $N_{123}$ due to its connection to the ground state |0,1/2,0〉, while the latter dome has the smaller tripartite negativity $N_{123}$ as it appears above the ground state |1,3/2,1〉. It is worth mentioning that the tripartite negativity $N_{123}$ also exhibits a qualitatively similar dependence for the particular case $J_{1}/J=1.0$ except that it becomes somewhat stronger in a range of moderate magnetic fields h/J∈(1,2) due to the coexistence of the ground states |1,3/2,1〉 and |1,1/2,1〉 (see Figure 6c). Finally, the highest tripartite entanglement between the peripheral spins can be found in the ground state |1,1/2,1〉 emergent at a higher value of the interaction ratio $J_{1}/J=1.25$, which is manifested in a range of moderate magnetic fields by the strongest tripartite negativity $N_{123}$ persistent up to relatively high temperatures $k_{B}T/J\approx 1.0$ (see Figure 6d).

## 4. Conclusions

In the present article we have examined in detail the spatial distribution of the bipartite and tripartite entanglement of the spin-1/2 Heisenberg star in the presence of a magnetic field using the Kambe projection method, which allows a straightforward computation of all eigenvalues and eigenvectors. The bipartite and tripartite entanglement of the spin-1/2 Heisenberg star was quantified through the bipartite and tripartite negativity, which was analytically calculated as the sum of the absolute values of all negative eigenvalues of the partially transposed reduced density matrix. Except for a trivial separable state with fully saturated spins, the spin-1/2 Heisenberg star additionally displays three nonseparable ground states with pronounced bipartite and tripartite quantum entanglement.

It has been found that the ground state |1,3/2,1〉 exhibits bipartite and tripartite entanglement over all possible decompositions of the spin-1/2 Heisenberg star into any pair or triad of spins. However, the bipartite and tripartite entanglement between the peripheral spins is much weaker in comparison to the bipartite and tripartite entanglement between the central and peripheral spins. An even more paradoxical situation emerges within the ground state |0,1/2,0〉, which contrarily exhibits tripartite entanglement between any triad of spins in spite of the complete lack of bipartite entanglement. The central spin of the spin-1/2 Heisenberg star is separable from the remaining three peripheral spins within the ground state |1,1/2,1〉, which accordingly exhibits just bipartite and tripartite entanglement between the peripheral spins related to a two-fold degenerate W-state. The two-fold degeneracy of the ground state |1,1/2,1〉 is responsible for a reduction in the tripartite negativity to half of the value, which is generally expected for the W-state. In spite of this fact, the tripartite thermal entanglement between three peripheral spins within the W-state |1,1/2,1〉 of the spin-1/2 Heisenberg star turns out to be most the robust against rising temperature and hence, this quantum state is most favorable for quantum computation.

## Figures and Tables

**Figure 1 molecules-28-04037-f001:**
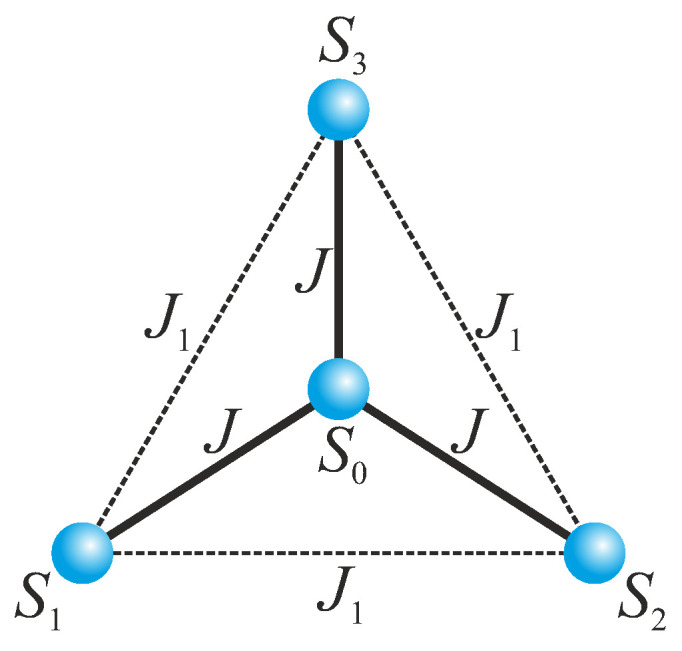
A schematic illustration of the magnetic structure of the spin-1/2 Heisenberg star composed from the central spin $S_{0}$ and three peripheral spins $S_{1}$, $S_{2}$ and $S_{3}$. Solid and broken lines denote the coupling constants *J* and $J_{1}$ ascribed to two different exchange interactions.

**Figure 2 molecules-28-04037-f002:**
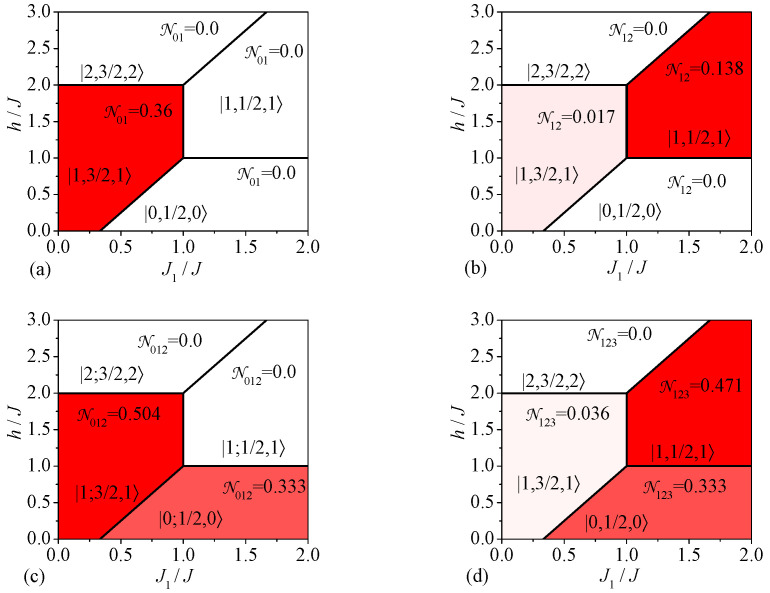
Zero-temperature density plots of the bipartite and tripartite negativities serving as ground-state phase diagrams of the spin-1/2 Heisenberg star: (**a**) the bipartite negativity $N_{01}$; (**b**) the bipartite negativity $N_{12}$; (**c**) the tripartite negativity $N_{012}$; (**d**) the tripartite negativity $N_{123}$.

**Figure 3 molecules-28-04037-f003:**
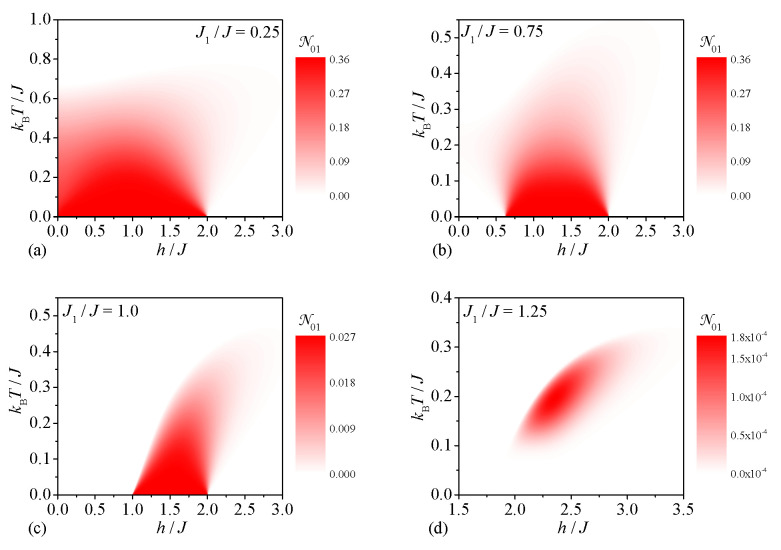
The bipartite negativity $N_{01}$ between the central and peripheral spins as a function of the magnetic field and temperature for four different values of the interaction ratio: (**a**) $J_{1}/J=0.25$, (**b**) $J_{1}/J=0.75$, (**c**) $J_{1}/J=1.0$, (**d**) $J_{1}/J=1.25$.

**Figure 4 molecules-28-04037-f004:**
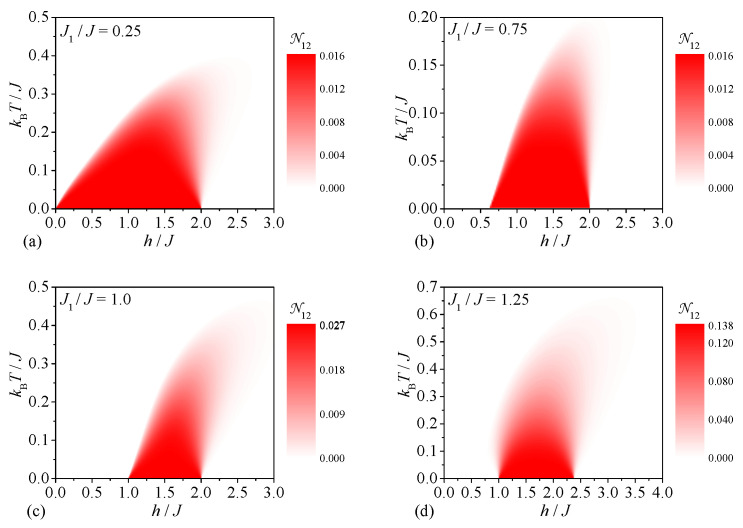
The bipartite negativity $N_{12}$ between two peripheral spins of the spin-1/2 Heisenberg star as a function of the magnetic field and temperature for four different values of the interaction ratio: (**a**) $J_{1}/J=0.25$, (**b**) $J_{1}/J=0.75$, (**c**) $J_{1}/J=1.0$, (**d**) $J_{1}/J=1.25$.

**Figure 5 molecules-28-04037-f005:**
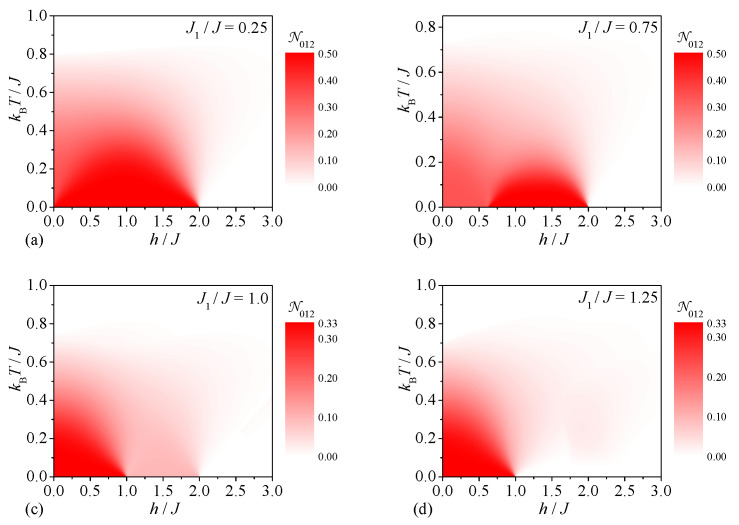
The tripartite negativity $N_{012}$ between the central and two peripheral spins as a function of the magnetic field and temperature for four different values of the interaction ratio: (**a**) $J_{1}/J=0.25$, (**b**) $J_{1}/J=0.75$, (**c**) $J_{1}/J=1.0$, (**d**) $J_{1}/J=1.25$.

**Figure 6 molecules-28-04037-f006:**
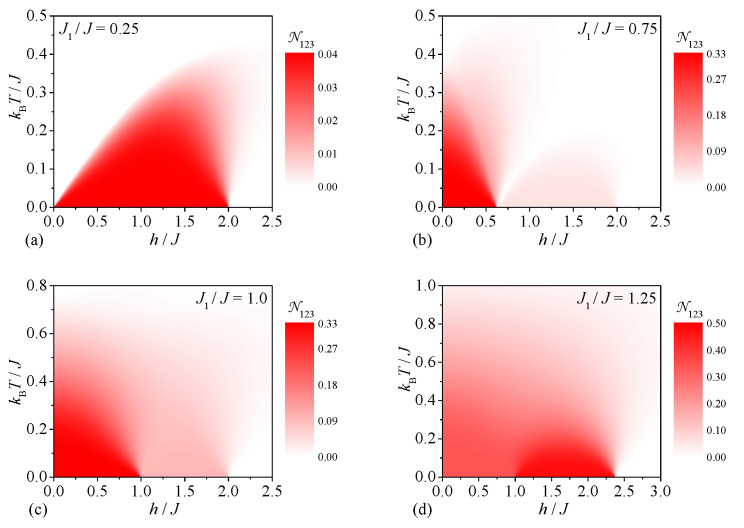
The tripartite negativity $N_{123}$ between the three peripheral spins as a function of the magnetic field and temperature for four different values of the interaction ratio: (**a**) $J_{1}/J=0.25$, (**b**) $J_{1}/J=0.75$, (**c**) $J_{1}/J=1.0$, (**d**) $J_{1}/J=1.25$.

**Table 1 molecules-28-04037-t001:** Eigenvalues and eigenvectors of a spin-1/2 Heisenberg star in a magnetic field given by the Hamiltonian (Equation 1). Arrows express the *z*-component of the spins. For instance, the eigenvector |↑↑↓↑〉 corresponds to the particular state with the following spin orientation: $S_{0}^{z}=1/2,S_{1}^{z}=1/2,S_{2}^{z}=-1/2,S_{3}^{z}=1/2$.

$|S_{T},S_{▵},S_{T}^{z}\rangle$	Eigenergies	Eigenvectors
|2,3/2,2〉	$E_{1}=\frac{3}{4}J+\frac{3}{4}J_{1}-2h$	$|\psi_{1}\rangle \;=|\;↑↑↑↑\rangle$
|2,3/2,−2〉	$E_{2}=\frac{3}{4}J+\frac{3}{4}J_{1}+2h$	$|\psi_{2}\rangle \;=|\;↓↓↓↓\rangle$
|2,3/2,1〉	$E_{3}=\frac{3}{4}J+\frac{3}{4}J_{1}-h$	$|\psi_{3}\rangle \;=\frac{1}{2}(|\;↓↑↑↑\rangle +|\;↑↓↑↑\rangle +|\;↑↑↓↑\rangle +\left|\;↑↑↑↓\rangle \right)$
|2,3/2,−1〉	$E_{4}=\frac{3}{4}J+\frac{3}{4}J_{1}+h$	$|\psi_{4}\rangle \;=\frac{1}{2}(|\;↑↓↓↓\rangle +|\;↓↑↓↓\rangle +|\;↓↓↑↓\rangle +\left|\;↓↓↓↑\rangle \right)$
|2,3/2,0〉	$E_{5}=\frac{3}{4}J+\frac{3}{4}J_{1}$	$|\psi_{5}\rangle \;=\frac{1}{\sqrt{6}}(|\;↓↑↑↓\rangle +|\;↓↑↓↑\rangle +|\;↓↓↑↑\rangle +|\;↑↓↑↓\rangle$
		+ |↑↑↓↓〉+|↑↓↓↑〉)
|1,3/2,1〉	$E_{6}=-\frac{5}{4}J+\frac{3}{4}J_{1}-h$	$|\psi_{6}\rangle \;=\frac{\sqrt{3}}{2}|\;↓↑↑↑\rangle -\frac{\sqrt{3}}{6}(|\;↑↓↑↑\rangle +|\;↑↑↓↑\rangle +\left|\;↑↑↑↓\rangle \right)$
|1,3/2,−1〉	$E_{7}=-\frac{5}{4}J+\frac{3}{4}J_{1}+h$	$|\psi_{7}\rangle \;=\frac{\sqrt{3}}{2}|\;↑↓↓↓\rangle -\frac{\sqrt{3}}{6}(|\;↓↑↓↓\rangle +|\;↓↓↑↓\rangle +\left|\;↓↓↓↑\rangle \right)$
|1,3/2,0〉	$E_{8}=-\frac{5}{4}J+\frac{3}{4}J_{1}$	$|\psi_{8}\rangle \;=\frac{1}{\sqrt{6}}(|\;↓↑↑↓\rangle +|\;↓↑↓↑\rangle +|\;↓↓↑↑\rangle -|\;↑↓↑↓\rangle$
		− |↑↑↓↓〉−|↑↓↓↑〉)
|1,1/2,1〉	$E_{9}=\frac{1}{4}J-\frac{3}{4}J_{1}-h$	$|\psi_{9}\rangle \;=\frac{1}{\sqrt{3}}(|\;↑↑↑↓\rangle +\exp\left(\frac{i2\pi }{3}\right)|\;↑↑↓↑\rangle$
		$+\exp\;(\frac{i4\pi }{3})|\;↑↓↑↑\rangle )$
|1,1/2,1〉	$E_{10}=\frac{1}{4}J-\frac{3}{4}J_{1}-h$	$|\psi_{10}\rangle \;=\frac{1}{\sqrt{3}}(|\;↑↑↑↓\rangle +\exp\left(\frac{i4\pi }{3}\right)|\;↑↑↓↑\rangle$
		$+\exp\;(\frac{i2\pi }{3})|\;↑↓↑↑\rangle )$
|1,1/2,−1〉	$E_{11}=\frac{1}{4}J-\frac{3}{4}J_{1}+h$	$|\psi_{11}\rangle \;=\frac{1}{\sqrt{3}}(|\;↓↓↓↑\rangle +\exp\left(\frac{i2\pi }{3}\right)|\;↓↓↑↓\rangle$
		$+\;\exp(\frac{i4\pi }{3})|\;↓↑↓↓\rangle )$
|1,1/2,−1〉	$E_{12}=\frac{1}{4}J-\frac{3}{4}J_{1}+h$	$|\psi_{12}\rangle \;=\frac{1}{\sqrt{3}}(|\;↓↓↓↑\rangle +\exp\left(\frac{i4\pi }{3}\right)|\;↓↓↑↓\rangle$
		$+\exp(\frac{i2\pi }{3})|\;↓↑↓↓\rangle )$
|1,1/2,0〉	$E_{13}=\frac{1}{4}J-\frac{3}{4}J_{1}$	$|\psi_{13}\rangle \;=\frac{1}{\sqrt{6}}(|\;↑↓↓↑\rangle +\exp\left(\frac{i2\pi }{3}\right)|\;↑↓↑↓\rangle$
		$+\;\exp\left(\frac{i4\pi }{3}\right)|\;↑↑↓↓\rangle -|\;↓↑↑↓\rangle -\exp\left(\frac{i2\pi }{3}\right)|\;↓↑↓↑\rangle$
		$-\;\exp(\frac{i4\pi }{3})|\;↓↓↑↑\rangle )$
|1,1/2,0〉	$E_{14}=\frac{1}{4}J-\frac{3}{4}J_{1}$	$|\psi_{14}\rangle \;=\frac{1}{\sqrt{6}}(|\;↑↓↓↑\rangle +\exp\left(\frac{i4\pi }{3}\right)|\;↑↓↑↓\rangle$
		$+\exp\left(\frac{i2\pi }{3}\right)|\;↑↑↓↓\rangle -|\;↓↑↑↓\rangle -\exp\left(\frac{i4\pi }{3}\right)|\;↓↑↓↑\rangle$
		$-\exp(\frac{i2\pi }{3})|\;↓↓↑↑\rangle )$
|0,1/2,0〉	$E_{15}=-\frac{3}{4}J-\frac{3}{4}J_{1}$	$|\psi_{15}\rangle \;=\frac{1}{\sqrt{6}}(|\;↑↓↓↑\rangle +\exp\left(\frac{i2\pi }{3}\right)|\;↑↓↑↓\rangle$
		$+\exp\left(\frac{i4\pi }{3}\right)|\;↑↑↓↓\rangle +|\;↓↑↑↓\rangle +\exp\left(\frac{i2\pi }{3}\right)|\;↓↑↓↑\rangle$
		$+\exp(\frac{i4\pi }{3})|\;↓↓↑↑\rangle )$
|0,1/2,0〉	$E_{16}=-\frac{3}{4}J-\frac{3}{4}J_{1}$	$|\psi_{16}\rangle \;=\frac{1}{\sqrt{6}}(|\;↑↓↓↑\rangle +\exp\left(\frac{i4\pi }{3}\right)|\;↑↓↑↓\rangle$
		$+\exp\left(\frac{i2\pi }{3}\right)|\;↑↑↓↓\rangle +|\;↓↑↑↓\rangle +\exp\left(\frac{i4\pi }{3}\right)|\;↓↑↓↑\rangle$
		$+\exp(\frac{i2\pi }{3})|\;↓↓↑↑\rangle )$

## Data Availability

Not applicable.

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
