# Peer review of "Distribution of Bipartite and Tripartite Entanglement within a Spin-1/2 Heisenberg Star in a Magnetic Field"

_molecules, 2023, doi:10.3390/molecules28104037_

Round 1

Reviewer 1 Report

The authors discussed the entanglement of a spin-1/2 Heisenberg star in a magnetic field. Both bipartite and tripartite entanglement are studied via the negativity. My comments and suggestions are as follows.

1) I think the authors should provide more discussions on why they choose negativity as a measure of entanglement here. It is known that there exist other tools for the quantification of entanglement such as the concurrence. Quantum Fisher information can also be used as a witness of entanglement. Hence, I think it is better to give proper motivation on the use of negativity, otherwise maybe calculating several tools and comparing their behaviors are a better choice. 

2) The notation of this paper is too heavy. There exist too many superscripts and subscripts. I suggest the authors find a way to simplify the notation so that the readers could have a better reading experience. 

The English writing is ok 

Author Response

Thank you very much for the positive evaluation of our manuscript. Below is enclosed our reply to reviewer comments:

  1. We have added to the Introduction the sentence on lines 55-58:
    "The main advantage of the quantity negativity with respect to other entanglement measures and witnesses lies in that it can be relatively simply calculated as a measure of both bipartite as well as multipartite entanglement [23–27]." which clarifies why we have used as a measure of entanglement negativity and not any other entanglement measure.
  2. Please know that the simplification of the notation would be at the cost of physical understanding, namely, two indices are always unambiguously determine the matrix elements, while other two (or three) indices specify pair (or triad) of spins for which the relevant element was calculated. From this perspective, there does not exist any simpler notation.

Reviewer 2 Report

The manuscript “Distribution of Bipartite and Tripartite Entanglement within a Spin-1/2 Heisenberg Star in a Magnetic Field" is a research of the spatial distribution of entanglement within the spin-1/2 Heisenberg star. Logical, detailed and understandable calculations for the spatial distribution of bipartite and triangular entanglement are given. The work makes a good impression, is written in a clear language and, in general, does not need edits.

I think, this manuscript can be published in the Molecules in present form.

Author Response

Thank you very much for your positive evaluation of our manuscript and suggestion to publish the paper in its present form.